# A qualitative study of stressors faced by older stroke patients in a convalescent rehabilitation hospital

Yuta Asada[1]*, Kaori Nishio[2], Kohei Iitsuka[3], Jun Yaeda[4]

1 Department of Occupational Therapy, Tokyo Bay Rehabilitation Hospital, Narashino, Chiba, Japan,
2 Department of Occupational Therapy, Teikyo Heisei University, Toshima, Tokyo, Japan, 3 Department of Occupational Therapy, Saitama Medical Center, Kawagoe, Saitama, Japan, 4 Faculty of Human Sciences, Graduate School of Comprehensive Human Sciences, University of Tsukuba, Ibaraki, Japan

* asaaaada10@gmail.com

**Data Availability Statement:** All relevant data are within the manuscript and its Supporting Information files.

**Funding:** The author(s) received no specific funding for this work.

## Abstract

This study aimed to explore the stressors experienced by older patients with stroke in convalescent rehabilitation wards in Japan. Semi-structured interviews were conducted with four stroke patients aged > 65 years who experienced a stroke for the first time in their lives. The interviews were analyzed using the Steps for Coding and Theorization method for qualitative data analysis. The results of the qualitative analysis demonstrated that patients experienced specific stressors, such as, difficulty in movement of the paralyzed hand, fear of stroke recurrence, and dietary problems. Some stressors were manageable through healthcare professionals' active and sensitive communication strategies. These stressors were derived from the theoretical framework of "stressors related to hospitalization" and "stressors related to the illness". Additional stressors emerged from the interaction between these two types within the theoretical framework. The results of this study contribute to a deeper understanding of the specific stressors experienced by older stroke patients during the recovery process.

## Introduction

Stress is a nonspecific response of the body to external stimuli [1]. Stress varies as the stressors faced by individuals differ depending on their age, sex, and social role [2]. Stressors include physical, biological, chemical, psychological, and social factors. The accumulation of these stressors causes stress, which, if not adequately addressed, can lead to physical or mental health problems, such as cardiovascular disease and depression, respectively [3]. To prevent these stress-related diseases, it is imperative to identify and address the stressors.

Patients often face various stressors in inpatient settings as their physical and human environments differ significantly from those of their regular home settings [4]. As the length of the hospital stay increases, patients may become particularly vulnerable to stressors such as "concern for family" and "anxiety about financial situation" [4]. The severity of a stroke, the age of the patient, and the presence of underlying medical conditions are factors that tend to extend

**Competing interests:** The authors have declared that no competing interests exist.

the duration of hospitalization [5]. The incidence of stroke increases with age and is more common among older adults [6]. Moreover, patients present with a variety of symptoms, such as motor paralysis and higher brain dysfunction, and their ability to perform activities of daily living (ADL) becomes more limited. In particular, convalescent rehabilitation hospitals have a prolonged hospital stay [7] as one of their goals is to help patients return to the community and their homes.

Much of what is known about stressors related to stroke involves the risk of stroke onset [8, 9], and there are insufficient studies on the stressors faced by older stroke patients in hospitals. Clarifying these unspoken stressors can contribute to reducing the stress of hospitalization for older stroke patients during convalescent rehabilitation, meeting their true needs, and enriching their lives after discharge. Few studies have elicited patients' true feelings regarding stressors in convalescent rehabilitation wards. The purpose of this study is to provide a deeper understanding of the specific stressors experienced by older stroke patients in convalescent rehabilitation wards during their hospital stay.

## Materials and methods

We conducted a qualitative study and interviewed each participant separately. The interview transcripts were analyzed according to the "Steps for Coding and Theorization" method (SCAT), a sequential and thematic qualitative data analysis technique [10–12].

This study was conducted in accordance with the Consolidated Criteria for Reporting Qualitative Research (COREQ), a checklist designed to improve the transparency and reliability of qualitative research [13] (S1 Table in S1 File).

### Preparation for the study

The first author (hereafter, "the author") is a M.S. student in comprehensive human sciences and male occupational therapist with six years of clinical experience in recovery rehabilitation. Before this study was conducted, the author reviewed the literature on SCAT, conducted an analysis, and attended a workshop for SCAT developers to deepen his understanding of the analysis methods to ensure the accuracy of the analysis [10–12].

### Participants

Patients aged 65 years or older, experiencing stroke for the first time, and hospitalized in a recovery center were included in the study. Patients who had difficulty answering the interview questions owing to the effects of aphasia, hospitalized patients in the charge of an interviewer, patients diagnosed with dementia or psychiatric disorders, and patients who were hospitalized for a short period of approximately one month were excluded.

Patients were asked to cooperate in the study and fully informed about the purpose and significance of the study, research methods, voluntary nature of research cooperation and freedom to withdraw, and handling of personal information. Signing a consent form indicated patients' willingness to cooperate in the study.

### Interview procedure

Three interviews were conducted between June and November, 2022. The interviewer asked questions according to an interview guide. Semi-structured in-person interviews were conducted in a private room in the hospital that the author is affiliated with, involving the patients and interviewer only. The first interview was conducted at the time of hospital admission, and subsequent interviews were conducted several times, with a gap of approximately one month.

The interviews were recorded with the participants' consent using the voice recorder function of an iPad and transcribed afterwards. The interview transcripts were not returned to participants for comments or correction. The interviewer recited the patients' statements to them and made efforts to confirm the content of the statements to ensure data accuracy.

The interview guide was developed based on a preliminary survey of two stroke patients to determine ease of response. The content of the interview guide was first explained to the participants through specific examples to help them fully understand the difference between "stress" and "stressors." The guide began by explaining, through specific examples, what the stressors in this study were. To investigate the stressors faced by older stroke patients in recovery, we asked, "What comes to mind when you hear the term 'stressors in hospitalization'?"

## Data analysis

We predicted that the outcome of the interviews would be strongly influenced by the participants' individual characteristics. Therefore, to obtain objective results, we used the SCAT technique that specializes in coding and theorization and can be applied to a small amount of data. The SCAT method consists of the following steps [10–12]:

Step 1: Focus words from within the interview texts.

Step 2: Words outside the text that can replace the words from Step 1.

Step 3: Words that explain the words in Step 1 and Step 2.

Step 4: Themes and constructs, including the process of writing a story and offering theories that weave the themes and constructs together.

As this study was designed to create multiple storylines from a single participant, we integrated those multiple storylines into a single storyline and wrote a theoretical description, ensuring no loss of chronological contextuality and individuality of the storylines. The data analysis and confirmation process were conducted by the author and three other authors who were not involved in the interview process.

## Ethical considerations

This study was approved by the Ethical Review Committee (Approval No. 289–2) of Tokyo Bay Rehabilitation Hospital.

## Results

### Basic attributes of the participants

Five participants who met the inclusion criteria were recruited for the study. One participant (female) was excluded owing to early discharge from the hospital on short notice. Thus, four patients (two male and two female) were included in the study. The participants' average age was 79.3 years (range: 71–88 years). Their disabilities included cerebral hemorrhage (one patient) and cerebral infarctions (three patients).

### Interview procedure

The average duration of the series of 12 interviews was 20.3 minutes, ranging from 7.5 to 32.7 minutes.

### Storyline and theoretical descriptions

In the sections below, the storylines and theoretical descriptions as well as quotes from each participant, are described.

**Case 1: Mr. A, facing an inconvenient situation.** At the time of the first interview, Mr. A experienced stress owing to an inconvenient situation during hospitalization. He was unable to perform the activities he did before the onset of the disease, especially owing to the psychological burden caused by the inability to eat and drink according to his preferences. He also expressed dissatisfaction with the current situation, limitations in leisure-time activities, inconvenience of activities, and a sense of shame caused by assistance with bathing. Limited leisure-time activities resulted from challenges in moving his paralyzed hands. He specifically encountered difficulties in willingly engaging them to act. Furthermore, he was separated from his family as a result of hospitalization. Thus, he faced restrictions in eating and drinking luxury foods, lack of freedom in daily life, and lack of family time.

"*Not being able to do things freely is the biggest stressor. All in all, there's nothing better than that. I can't eat what I like, or drink a lot. Even if I have a computer, I can't use my right hand. I can't even do my own hobbies. And, it is still significant whether or not you have a wife nearby.*"

At the time of the second interview, Mr. A experienced stress regarding eating and drinking, including dissatisfaction with the variety of meals compared to before the disease onset, and the psychological burden owing to meals not being replaced on a daily basis. This was also the minimum element that Mr. A looked for during hospitalization. Other stress factors included a feeling of disappointment owing to limited leisure-time activities, and feelings of activity limitation and resignation owing to the inability to walk independently.

"*The most important thing is the food. Anyway, there's nothing to do, so at least a meal, you'd think, wouldn't you? The food is different from when we're at home. It doesn't help that I can't walk. And, I think it's a bit hard not to have hobbies.*"

At the time of the third interview, Mr. A expressed that his biggest stress factor was difficulty moving his paralyzed dominant hand. This significantly impacted his daily self-care, including toileting and grooming. He also encountered limitations in various leisure activities, such as reading books. Eating and drinking induced a significant psychological burden. He felt dissatisfied with the lack of variety in meals as he could not manage to eat as well as previously.

"*Whatever I do, my hands don't work. For example, when you brush your teeth. It's the same when you go to the toilet and wipe your bottom. I can't use my right hand. Also, I like books and I want to read, but I can't turn the pages. And, unlike in the past, I eat rice and side dishes every day. My eating habits have changed drastically.*"

**Case 2: Ms. B, facing stressors caused by communal living.** At the time of the first interview, Ms. B faced stressors related to basic lifestyle habits, such as falling asleep and toileting, in the hospital. Variations in individual lifestyles and environmental factors, like noise and room brightness, contributed to sleep deprivation in shared living arrangements. Furthermore, inadequate management of the paralyzed side during sleep led to anxiety and sleep deprivation. Problems related to toileting needs arose owing to overlap in toilet timings with roommates and assisted by staff of the opposite sex.

"*I sometimes have trouble sleeping well at night because of noises or brightness. Everyone is trying to go to the toilet before rehabilitation, so the timing is. . . And with male nurses, there*

*was a bit of resistance to using the toilet. After all, in shared living arrangements, everyone has a different rhythm of life."*

During the second interview, Ms. B continued to face stress owing to communal living. Stressors included abnormal breathing noises caused by roommates when falling asleep, noise problems during roommates' movements, and nocturnal awakenings caused by physical environmental factors such as differences in depth of sleep. Additionally, there were case of sleep problems caused by the staff's response to a roommate's problematic behavior, and case of nocturnal awakenings caused by noise from staff responses. Other issues included self-perceived persistent distress over defecation problems and dealing with defecation needs in a time-constrained environment, with a roommate.

*"Like last time, in shared living arrangements, everyone has a different rhythm of life, but it can't be helped. Sleep, you know, because some people go to the toilet at night or early in the morning, so it's quite noisy and you can't sleep well. And the nurse puts the patient next to me to sleep, and there are all sorts of noises when she does that. We all have the same desire to go to the toilet before rehabilitation, so we don't make it in time. Toilets are a perpetual problem."*

At the time of the third interview, Ms. B had problems with how he interacted with his roommates and stressors related to falling asleep at night. Ms. B was dissatisfied with differences in personal characteristics in communal living, and concerned about the deterioration of his relationship with his roommates over defecation. Furthermore, stress was caused by differences in lifestyle in communal living affecting sleep and awakening during the night owing to physical environmental factors such as noises made by roommates. Sudden changes in training hours also caused dissatisfaction.

*"Like how to communicate with people in the room. Like sleeping. Because of the lights and noise when my roommate goes to the toilet at night. Roommates have different living patterns. In rehabilitation, though, there were some questionable things like time changes."*

**Case 3: Ms. C, facing an excrement problem and anxiety about stroke recurrence.** At the time of the first interview, Ms. C faced the problem of excrement in communal living. Dissatisfaction was caused by the suppression of excretory behavior and rejection of excretion in communal living, leading to anxiety. There were also conflicts and a psychological burden caused by the staff's lack of information sharing, which led to restraining from defecating after unpleasant experiences.

*"I don't like the situation of one toilet for four people. I and others are suffering. I thought it was hard. I didn't know that you have to press the nurse call. Then I wished they had told me from the beginning. That was a bit of a shock."*

At the time of the second interview, Ms. C expressed dissatisfaction with their lack of independence in elimination. This led to a sense of aversion caused by dealing with the need to defecate frequently during the night and self-consciousness about requests for nighttime defecation assistance, which, in turn, led to resisting the need to defecate, a distressing experience unique to the patient.

"*I feel bad because I have to go to the toilet in the middle of the night. But I try to be patient. If it was during the day, I would ask the nurse to help me, but at night I would still feel sorry. It's painful. You have to be experienced to understand.*"

At the time of the third interview, Ms. C was anxious about the gap between their life at home after discharge and their life in the hospital and about the gradual decline of their brain functions. They also experienced anxiety owing to the fear of stroke recurrence and an undecided medical support system for the prevention of recurrence. These stressors were related to worry caused by a lack of information sharing by the staff and delays in sharing information about discharge from the hospital.

"*I have a little bit of anxiety about my future and my life. Because I've got comfortable here. And I don't know what I would do if I fell ill again. No one is going to talk to me about it. I'm a bit worried about that. That's what I'm most worried about.*"

**Case 4: Mr. D, facing a meal problem.** At the time of the first interview, Mr. D expressed their stress that they had to hold their toileting until the hospital staff arrived when they needed to defecate. This occurred as the hospital staff were extremely busy, and they experienced failure in excretory management. However, at the time of the interview, they were able to use the toilet independently.

"*I've had a leak before the nurse came. She can't come right away, she's too busy. It's gone now.*"

At the time of the second interview, Mr. D had a low appetite owing to low-temperature meals and refused to eat as a result of inappropriate meal temperature. Additionally, there were difficulties with grooming movements around the use of the wash basin and dealing with the need to defecate in communal living.

"*The rice and side dishes are cold. So I feel sorry to leave it. I can eat it beautifully when it's warm. But when it's cold, I just can't. After the meal, I can't wash my hands because some people wash their hands in their rooms first. When I want to go into the toilet, there are people ahead of me. It can't be helped.*"

In the third interview, Mr. D felt stress when the meal was not hot enough to eat and lost their appetite. He also felt stress when his mealtime was delayed as it that cause would take time for them to do their personal grooming after returning to their room where their roommate occupied t the wash basin.

"*Side dish is cold. Wish it was room temperature. I eat my meals late, so I'm the last one to go back to my room. So, I can't wash my hands first.*"

## Discussion

In this study, semi-structured interviews were conducted to identify the stressors faced by older patients with stroke during convalescent rehabilitation, throughout hospitalization; data analysis was conducted using SCAT.

Based on the storylines and theoretical descriptions, the stressors experienced by stroke patients were categorized into "stressors related to hospitalization" and "stressors related to the illness" [4].

## Stressors related to hospitalization

The results of this study revealed that older stroke patients in convalescent rehabilitation face stressors related to ADLs, such as eating, sleeping, grooming, and toileting; leisure activities; problems with roommates in communal living; and inability to be with their family members. In this study, the first interview was conducted at the time of admission, and stressors were reported by all participants. Stress during hospitalization is caused by the fact that patients are forced to live a life with less freedom than before [4].

The psychological burden is particularly high for older adults as they have a reduced ability to adapt to changes in the external environment compared with younger patients [14]. In light of the above, older stroke patients may face a variety of stressors from the early stages of hospitalization compared with younger patients; therefore, intervention against these stressors is necessary from the early stages of hospitalization.

Factors such as relationships with roommates may lead patients to experience discomfort [15], and the way patients relate to their roommates is considered important. In this study, physical environmental factors caused by differences in lifestyle and the timing of toilet and wash basin use with roommates emerged as stressors. Additionally, these factors affected the participants' ADL, such as grooming, toileting, and sleeping. Considering these findings, it is important for patients living together to consider each other's needs. Therefore, it is necessary for patients to communicate with each other to deepen their understanding, and healthcare professionals are expected to play a role in building such relationships.

Furthermore, stressors such as meal variations and meal temperature emerged rather than stressors such as taste and preference. Older people tend to experience a decline in dietary variety owing to a decline in physical and oral functions and appetite [16]. Moreover, older patients undergoing treatment for cerebrovascular disease are more likely to experience changes in food preferences than younger patients [17], which is not consistent with the results of the present study. Given that the amount of food intake in a hospital setting is linked to the quality of food, including taste and the dining environment [18, 19], there is a need for further research on qualitative aspects of meal preparation, such as food variations and appropriate temperatures. However, studies on meal variations and temperature are limited. In the future, these should be investigated in detail as characteristic stressors faced by older stroke patients during convalescent rehabilitation.

## Stressors related to the disease

The results revealed that older stroke patients in rehabilitation face stressors such as difficulty moving the hand affected by motor paralysis, recurrent strokes, lack of information given by healthcare providers, and inappropriate actions or words of healthcare providers. Approximately 50% of stroke survivors experience unilateral motor paralysis [20]. Improvement in motor paralysis of the upper limbs and fingers contributes to greater independence in ADL [21, 22]. It not only affects ADL but a wide range of activities, such as housework and leisure activities [23, 24].

In this study, there were patients whose hobbies were limited by difficulty in moving the paralyzed hand. Additionally, based on the interviews at the time of admission, activity limitation caused by paralysis was a stressor faced from the time of admission itself. Therefore, early interventions and psychological support are needed for patients with paralysis.

A lack of information about the disease may also increase patient anxiety and cause dissatisfaction among healthcare providers [4]. Stroke recurs at a rate of 2.2% to 25.4% within one year of disease onset, 12.9% within two years, and approximately 16% within five years [25]. Therefore, it is important to support stroke patients to prevent recurrence [26]. The participants

were interviewed before discharge from the hospital about stressors such as recurrent stroke and lack of information provided by healthcare providers. This suggests that providing information to older patients with stroke undergoing convalescent rehabilitation to prevent recurrence is very important, especially for patients who are about to be discharged from the hospital, and that a lack of information can cause stress. Furthermore, communication between stroke patients and healthcare professionals does not always match [27]. Efforts should be made to prevent a lack of information, considering the patient's cognitive function and the degree of higher brain dysfunction.

Additionally, stressors such as the personal care of patients by healthcare professionals of the opposite sex, and behaviors and words caused by misunderstandings on the part of healthcare professionals emerged. Patients may experience discomfort and high psychological distress owing to factors such as the attitudes and actions of healthcare workers [16, 28]. An inadequate explanation or lack of consideration of shame may also arouse anger in patients [29]. Stroke patients are placed in a situation where they are prone to feelings of shame owing to assistance with ADL such as bathing and toileting. Therefore, healthcare professionals must be sensitive to patients when providing daily care. Stress can be prevented through appropriate attitude and information sharing.

Various symptoms, such as motor paralysis, sensory disturbance, higher brain dysfunction, and cognitive decline, appear as post-effects of stroke. The complex interplay between these symptoms causes a decline in the ability to perform ADL [30–32]. In this study, there were patients for whom difficulty in achieving independence in ADL was a stressor. Patients with higher levels of ADL independence had higher self-efficacy, and successful experiences were effective in forming self-efficacy [33]. This principle should be applicable to older stroke patients in convalescent rehabilitation hospitals. The positive outcomes of their hospital experience may be partially attributed to reduced stress.

Additionally, some patients faced limitations in self-care, stressors related to hospitalization owing to the aftereffects of stroke, and stressors related to illness. Given these findings, it was suggested that stroke patients may have been stressed by the interaction of "stressors related to the disease" and "stressors related to hospitalization." However, if one of these stressors can be adequately addressed, it is likely that related stressors can be reduced.

## Limitations

In conclusion, we clarified the stressors faced by older stroke patients in convalescent for rehabilitation. However, this study has some limitations. First, the study was severely limited by the small number of patients, which prevents us from drawing some important conclusions. The SCAT method can be used to analyze data from a small number of people because it provides a theoretical description from the participants' storylines; however, the number of participants in this study was not sufficient to generalize the findings. Second, this study did not fully consider the participants' individual characteristics, such as personality and background, nor did it analyze the patients in terms of their pathology and sequelae. Therefore, the results obtained should be interpreted carefully, as individual bias was not sufficiently eliminated. In future, it is necessary to select other participants and data analysis methods that consider participants' individual characteristics and the aftereffects of stroke and recruit more participants to elucidate the stressors faced by older stroke patients in convalescent rehabilitation.

## Conclusion

Stressors specific to older stroke patients were identified, including difficulty moving the paralyzed hand, recurrent stroke, and diet-related stressors. Stressors identified in this study can be

broadly classified into "stressors related to hospitalization" and "stressors related to the disease," consistent with previous studies [4]. However, it was found that stress is also caused by the interaction between "stressors related to hospitalization" and "stressors related to the disease." To the best of our knowledge, thus far, no reports have identified the specific stressors faced by older stroke patients. Therefore, this study provides valuable information from a first-hand perspective that will lead to a deeper understanding of the specific stressors experienced by older stroke patients during recovery. Future studies should explore how various stressors lead to stress in older stroke patients at various types of rehabilitation hospitals.

## Supporting information

**S1 File. Consolidated criteria for reporting qualitative studies (COREQ): a 32-item checklist.**
(DOCX)

## Acknowledgments

We thank all the participants who agreed to be interviewed for this study. We also thank the members of the Rehabilitation Science Degree Program, Graduate School of Comprehensive Human Sciences, University of Tsukuba, for their guidance and encouragement during this study.

## Author Contributions

**Formal analysis:** Yuta Asada.

**Investigation:** Yuta Asada.

**Methodology:** Yuta Asada.

**Writing – original draft:** Yuta Asada.

**Writing – review & editing:** Kaori Nishio, Kohei Iitsuka, Jun Yaeda.

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
