## [Decision Letter · Decision Letter 0]

14 Jun 2024

PONE-D-24-02744A qualitative study of stressors faced by older stroke patients in a convalescent rehabilitation hospitalPLOS ONE

Dear Dr. Asada,

Thank you for submitting your manuscript to PLOS ONE. After careful consideration, we feel that it has merit but does not fully meet PLOS ONE’s publication criteria as it currently stands. Therefore, we invite you to submit a revised version of the manuscript that addresses the points raised during the review process.

We look forward to receiving your revised manuscript.

Kind regards,

Chinh Quoc Luong, MD., PhD.

Academic Editor

PLOS ONE

Journal Requirements:

2. PLOS requires an ORCID iD for the corresponding author in Editorial Manager on papers submitted after December 6th, 2016. Please ensure that you have an ORCID iD and that it is validated in Editorial Manager. To do this, go to ‘Update my Information’ (in the upper left-hand corner of the main menu), and click on the Fetch/Validate link next to the ORCID field. This will take you to the ORCID site and allow you to create a new iD or authenticate a pre-existing iD in Editorial Manager. Please see the following video for instructions on linking an ORCID iD to your Editorial Manager account: https://www.youtube.com/watch?v=_xcclfuvtxQ".

Reviewers' comments:

Reviewer's Responses to Questions

**Comments to the Author**

1. Is the manuscript technically sound, and do the data support the conclusions?

Reviewer #1: No

Reviewer #2: Partly

2. Has the statistical analysis been performed appropriately and rigorously? 

Reviewer #1: N/A

Reviewer #2: N/A

3. Have the authors made all data underlying the findings in their manuscript fully available?

Reviewer #1: No

Reviewer #2: No

4. Is the manuscript presented in an intelligible fashion and written in standard English?

Reviewer #1: Yes

Reviewer #2: Yes

5. Review Comments to the Author

Reviewer #1: I found that this manuscript needs major revisions for publication in PLOSONE. This manuscript should follow the Consolidated Criteria for Reporting Qualitative Research (COREQ): a 32-item checklist for interviews and focus groups.

Reviewer #2: I want to congratulate the authors with this interesting study. However, the are several concerns that need be addressed before publication.

Abstract: The authors state: This study yields valuable insights that contribute to a deeper 33 comprehension of the specific stressors experienced by older patients with stroke during the recovery process. It is however unclear what these insights are. It would be valuable to give some insights in the results of this study so clinicians can decide if this study is useful for them

Introduction line 40: please elaborate on the mental health problems.

Introduction line 41: Which adverse effects do you mean? It is important to be a bit more specific.

Line 58: please state the aim of the study more clearly

Page 5 line 61: start with describing the design of the study.

Page 5 line 67: please provide a reference for SCAT

Page 6 line 92: I think a strength of this study is the methodological approach (use of SCAT). It might be helpful to also state this in the abstract. The use of this approach might justify the low sample size.

Results: The results section is very short. The authors should provide more demographic information about the participants and also report quotes from the participants.

Discussion: The results of this study are based on only 4 participants. Therefore, more information should be presented in the results section. These might also lead to a more profound discussion section. Also, the strengths and limitations should be described more thoroughly. Why did the authors chose this design and analysis? What are the consequences of the limited sample size etc.

6. PLOS authors have the option to publish the peer review history of their article (what does this mean?). If published, this will include your full peer review and any attached files.

Reviewer #1: No

Reviewer #2: No

---

## [Author Response · Author response to Decision Letter 0]

15 Jul 2024

Dear PLOS ONE

The authors thank you and the reviewers for your thoughtful suggestions and insights. 

Reviewer #1

We thank the reviewer for highlighting this issue. We have made revisions to the manuscript following the Consolidated Criteria for Reporting Qualitative Research (COREQ) guidelines. We request the reviewer to check the revised manuscript.

Reviewer #2

Thank you for the numerous suggestions. We realized that there were several points that needed clarification, including the purpose of the study, the reasons for adopting the analytical method, and the study limitations. Based on your feedback, we revised the manuscript.

Thank you for your consideration. I look forward to hearing from you.

Kind regards,

Yuta Asada

---

## [Editor Report · Decision Letter 1]

13 Aug 2024

A qualitative study of stressors faced by older stroke patients in a convalescent rehabilitation hospital

PONE-D-24-02744R1

Dear Dr. Asada,

We’re pleased to inform you that your manuscript has been judged scientifically suitable for publication and will be formally accepted for publication once it meets all outstanding technical requirements.

Kind regards,

Chinh Quoc Luong, MD., PhD.

Academic Editor

PLOS ONE

Additional Editor Comments (optional):

All comments have been addressed.
---

## [Editor Report · Acceptance letter]

15 Aug 2024

PONE-D-24-02744R1 

PLOS ONE

Dear Dr. Asada, 

I'm pleased to inform you that your manuscript has been deemed suitable for publication in PLOS ONE. Congratulations! Your manuscript is now being handed over to our production team.

Kind regards, 

on behalf of

Assoc. Prof. Chinh Quoc Luong 

Academic Editor

PLOS ONE